# Exploring the causal link between serum 25-hydroxyvitamin D concentrations and idiopathic sudden sensorineural hearing loss: Insights gained from a Mendelian randomization study involving two independent samples

Ying Zhao[1☯], Cong Yu[2☯], Hengchang Sun[3☯], Fangmei Xie[1], Jian Shen[1], Xiaoming Li[2], Xiaoyu Song[1], Wenfeng Luo[1], Jinhua He[1], Zeping Han[1]*

1 Central Laboratory, The Affiliated Panyu Central Hospital, Guangzhou Medical University, Guangzhou, Guangdong, P.R. China, 2 Clinical Laboratory Department, The Affiliated Panyu Central Hospital, Guangzhou Medical University, Guangzhou, Guangdong, P.R. China, 3 Department of Laboratory Medcine, Third Affiliated Hospital of Sun Yat-sen University, Guangzhou, Guangdong, P.R. China

☯ These authors contributed equally to this work.
* hanzeping1987@126.com

## Abstract

Idiopathic sudden sensorineural hearing loss (ISSNHL) is defined by the rapid onset of hearing impairment without an identifiable etiology. The decreased serum concentrations of 25-hydroxyvitamin D(25(OH)D) was shown to be associated with decreased hearing ability. However, current cross-sectional studies have only demonstrated an association, failing to establish a causal link. Therefore, a comprehensive investigation is necessary to clarify the causal relationship between them. Analysis was done by using largescale genome-wide association analysis(GWAS) summary datasets to give information about the incidence of ISSNHL and genetically predicted serum 25(OH)D levels by a two-sample Mendelian randomization (MR) analysis. Instrumental variables (IVs) were identified as genome-wide significant single-nucleotide polymorphisms (SNPs) with a significance threshold of $P < 1 \times 10^{-8}$ and an independence criterion of $r^2 < 0.001$. The GWAS data on serum 25-hydroxyvitamin D levels comprised 6,896,093 SNPs from 496,949 individuals of European ancestry (exposure variable). Outcome data were derived from another GWAS data, including 16,380,424 SNPs from 1,491 European ISSNHL cases and 196,592 controls. MR-Egger, inverse variance weighted (IVW), weighted median, simple mode, and weighted mode, were used to assess causal effects. Heterogeneity tests, horizontal pleiotropy tests, and the leave-one-out method were applied to evaluated the robustness of the MR analysis results. A total of 117 SNPs were employed as instrumental variables ($P < 5 \times 10^{-8}$). Our results indicated no causal association between serum 25(OH) D levels and the risk of ISSNHL within the European population (IVW; OR

**Data availability statement:** All relevant data are within the paper and its Supporting Information files.The exposure-related GWAS dataset for serum 25-hydroxyvitamin D levels, which included 496,946 samples of European descent and 6,896,093 SNPs, was retrieved from a publicly accessible repository, the IEU OpenGWAS Project (https://gwas.mrcieu.ac.uk, with the specific ID: ebi-a-GCST90000618). The ISSNHL GWAS summary data, comprising 16,380,424 SNPs from 1,491 cases of ISSNHL and 196,592 control individuals, were also obtained from the IEU OpenGWAS Project database (GWAS ID: finn-b-H8_HL_IDIOP), which included participants of European ancestry.

**Funding:** This study was supported by Basic and Applied Basic Research Foundation of Guangdong Province (No. 2022A1515220217; 2021A1515110533). The funders had no role in study design, data collection and analysis, decision to publish, or preparation of the manuscript.

**Competing interests:** The authors have declared that no competing interests exist.

= 1.09, 95% CI = 0.81 to 1.48, $P$ = 0.573). Furthermore, the statistical models did not reveal any evidence of heterogeneity or pleiotropy.

## Introduction

ISSNHL is characterized by an abrupt and unexplained decline in sensorineural hearing, often accompanied by symptoms such as tinnitus, vertigo, nausea, and vomiting, and it is increasingly recognized as a significant public health concern globally [1,2]. This condition predominantly affects individuals aged 40 and older, with an annual morbidity rate estimated between 5 to 20 cases per 100,000 individuals [3]. The prevalence of ISSNHL ranges from 0.07% to 5.2%, exhibiting an age-related increase and significant variability across different countries within the adult population [4]. There remains relatively scant information regarding the etiology of ISSNHL, thereby complicating the development of successful treatment strategies. Primary intervention for mitigating hearing loss involves early administration of repetitive systemic steroid therapy [5]. However, the absence of a definitive etiological diagnosis poses challenges for precise management of ISSNHL. Therefore, both prompt diagnosis and timely intervention are of utmost urgency and great significance. Recent research has embarked on exploring the potential role of vitamin D in ISSNHL, focusing on associated risk factors and underlying mechanisms, which presents a promising avenue for further investigation.

Vitamin D, a fat-soluble hormone predominantly synthesized through exposure to sunlight, can also be obtained from dietary sources and multivitamin supplements, which is critical for the maintenance of bone health and exhibits a range of beneficial effects, including antiproliferative and prodifferentiative actions, metabolic regulation, and support for neuromuscular function [6,7]. The potential influence of vitamin D on auditory health has attracted scholarly interest, with research indicating a link between vitamin D deficiency and hearing impairments. Evidence suggests that vitamin D is integral to the proliferation and differentiation of neural stem and progenitor cells, which are essential for proper functioning of the auditory nerve [8]. Vitamin D deficiency may result in cochlear calcium demineralization, potentially disrupting microcirculation, and leading to hearing loss [9]. Studies have identified a higher prevalence of vitamin D insufficiency among individuals with ISSNHL, suggesting a possible association between serum vitamin D level and the onset of ISSNHL [10]. This finding is further corroborated by Kose's research [11]. Furthermore, the role of vitamin D in the inner ear, potentially mediated through its receptors or its involvement in calcium metabolism, may contribute to sensorineural hearing loss when deficient [12]. Moreover, the influence of vitamin D on vascular cell function and its capacity to enhance the expression of anti-inflammatory cytokines suggests further mechanisms by which a deficiency could impact hearing [13,14]. However, traditional epidemiological interpretations may be complicated by various confounding factors, leading to the establishment of a causal relationship between the exposure factors and outcomes.

Mendelian randomization (MR) research design adheres to the Mendelian genetic law that "parental alleles are randomly allocated to offspring", offers a robust framework for evaluating the causal relationship between exposure and outcome by employing SNPs as IVs [15]. This methodological approach is inherently resistant to confounding factors, and can effectively avert the influence of confounding factors and eliminate the interference of reverse causality. As the genetic variations present in individuals are randomly assigned at conception, independent of environmental or behavioral influences. As a result, MR is often regarded as a natural counterpart to traditional randomized controlled trials (RCTs) [16]. In the present study, we sought to clarify the causal relationship between 25-hydroxyvitamin D levels, a recognized indicator of nutritional vitamin D status, and ISSNHL by utilizing MR methodology. To the best of our knowledge, this study represents an inaugural investigation of the causal link between serum 25-hydroxyvitamin D levels and the risk of ISSNHL through MR analysis.

## Methods and materials

### Study design

We conducted a two-sample MR study aimed at deriving a comprehensive and reliable conclusion regarding the causal relationship between serum 25-hydroxyvitamin D levels (the exposure) and ISSNHL (the outcome). Serum 25-hydroxyvitamin D levels can be assessed through blood tests. ISSNHL, which is typically diagnosed according to the "Guidelines for Diagnosis and Treatment of Sudden Deafness (2015)". A schematic representation of the study design is illustrated in Fig 1. This study is a bidirectional MR study, and follows three fundamental assumptions: Assumption 1: the selected instrumental variables exhibit a strong correlation with the exposure ($P < 5 \times 10^{-8}$); Assumption 2: there is no correlation between the IVs and the outcome; and Assumption 3: the IVs do not exert an influence on the outcome through confounding variables other than the exposure [17].

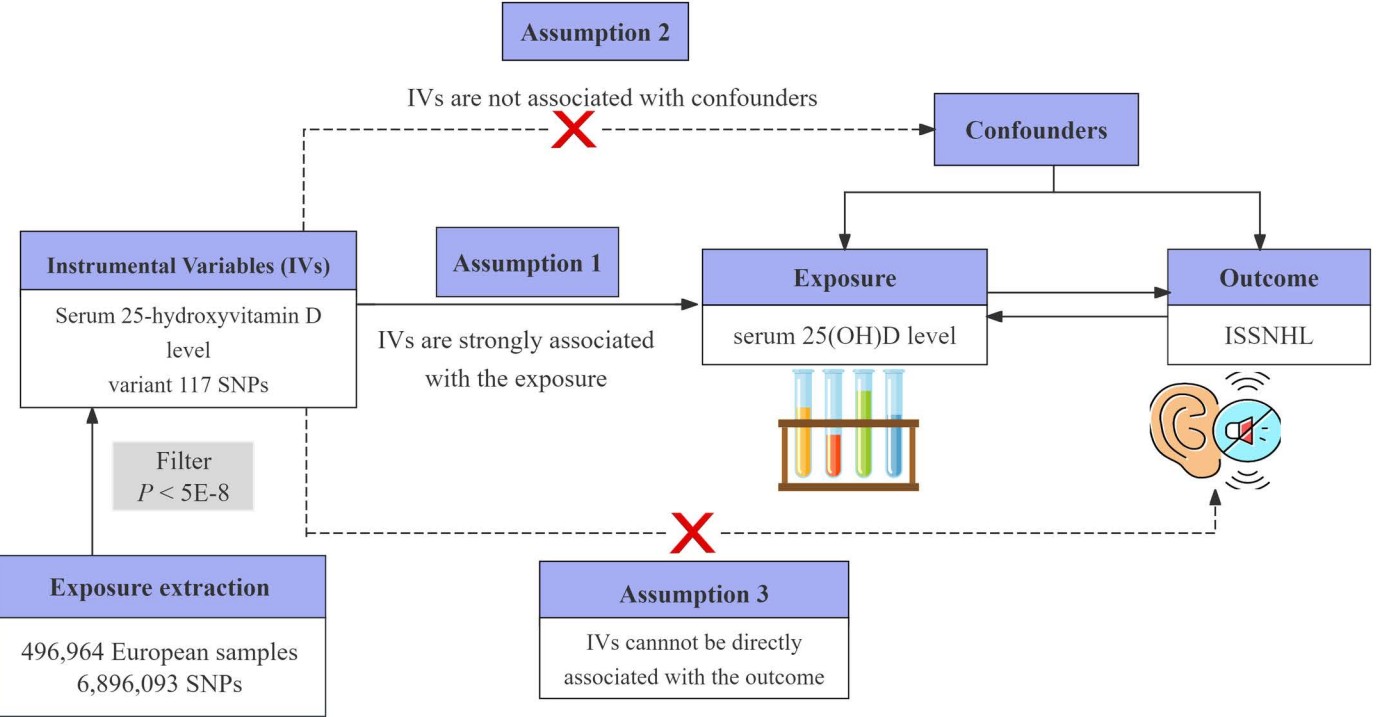

**Fig 1. Design of the MR study on the causal relationship between serum 25(OH)D levels and ISSNHL.**

## GWAS summary data for serum 25-hydroxyvitamin D levels and ISSNHL

Serum 25-hydroxyvitamin D levels were considered the exposure variable, with ISSNHL serving as the outcome variable. Data specifically related to serum 25-hydroxyvitamin D levels and ISSNHL were meticulously sourced from GWAS, ensuring that all individuals included in the study provided informed consent. The exposure-related GWAS dataset for serum 25-hydroxyvitamin D levels, which included 496,946 samples of European descent and 6,896,093 SNPs, was retrieved from a publicly accessible repository, the IEU OpenGWAS Project (https://gwas.mrcieu.ac.uk, with the specific ID: ebi-a-GCST90000618). ISSNHL GWAS summary data, comprising 16,380,424 SNPs from 1,491 cases of ISSNHL and 196,592 control individuals, were also obtained from the IEU OpenGWAS Project database (GWAS ID: finn-b-H8_HL_IDIOP), which included participants of European ancestry. Owing to the lack of individual-level data for the exposure and outcome samples, we were unable to assess potential overlaps between these samples.

## IVs selection

In accordance with the criteria established in our previous Mendelian randomization studies for SNPs selection [18], we ensured the independence of each SNP and minimized the potential impact of gene pleiotropy. SNPs that demonstrated a significant association with vitamin D levels were selected based on the following parameters: 1) A genome-wide significance threshold of $P < 5 \times 10^{-8}$ and a linkage disequilibrium (LD) measure of $r^2 < 0.001$ within a 10,000 kb window. 2) In order to mitigate bias arising from SNPs with weaker IVs, we calculated the F-statistics for each SNP using the formula: F-statistics = $(Beta/Se)^2$ [19]. 3) The resulting F-statistics values serve as indicators of the strength of the SNPs as instrumental variables, with a threshold exceeding 10 suggesting the absence of bias from weak IVs [20]. Potential IVs for serum 25-hydroxyvitamin D were systematically interrogated using both PhenoScanner V2 (http://www.phenoscanner.medschl.cam.ac.uk/) and LDlink's LDtrait module (https://ldlink.nih.gov/?tab=ldtrait), with critical confounders: osteoporosis, parathyroid hormone levels, coronary artery disease, hypertension, type 2 diabetes, and smoking behavior. SNPs demonstrating significant associations with these confounders were iteratively excluded to ensure the independence of IVs from known mediators or comorbidities of ISSNHL, thereby satisfying the key assumption of exclusion restriction in Mendelian randomization.

## MR analysis

Five distinct methodologies were utilized to investigate the causal relationship between vitamin D levels and ISSNHL. Including inverse variance weighting (IVW), MR-Egger regression, weighted median estimation (WME), weighted mode, and simple mode, with IVW being designated as the primary analytical technique [21,22]. The IVW method assigns weights to the SNP ratios of filtered SNPs, operating under the premise that if the effect size of the instrumental variable is zero, the corresponding outcome variable, namely the risk of ISSNHL, will also be zero. This approach presupposes the validity of all SNPs included in the MR analysis and aggregates the Wald ratios of each SNP to derive an overall weighted effect. The findings produced by the IVW method were regarded as the principal results of the study [23]. Additionally, the WME method was employed to derive a one-to-one estimate for each individual SNP by utilizing the median of the weight sizes.

Mendelian randomization-Egger (MR-Egger) represents a methodological approach within the framework of Mendelian randomization that utilizes summarized genetic data. The MR-Egger method allows us to assess whether genetic variants exert pleiotropic effects on the outcome that differ significantly from zero (directional pleiotropy). We employed the MR-PRESSO global test and MR-Egger regression to evaluate the pleiotropy of IVs, and $P < 0.05$ indicating the presence of pleiotropy.

## Causal interpretation and assumptions

The instrumental variable (IV) estimates derived from our MR analysis reflect the Local Average Treatment Effect (LATE) of serum 25-hydroxyvitamin D concentrations on ISSNHL. This effect specifically applies to "compliers"—individuals

whose vitamin D levels are influenced by the genetic variants used as instruments. To strengthen the plausibility of this assumption, we prioritized SNPs with genome-wide significant associations ($P < 5 \times 10^{-8}$) to serum 25-hydroxyvitamin D in prior GWAS, ensuring consistent directional effects across studies.

### Sensitivity analysis

The sensitivity analysis of the results was conducted using the leave-one-out method. If the remaining SNP results, after the removal of any single SNP, fall on the correct side of the invalid line, it indicates that the removal of that SNP does not affect the overall results. This approach helps to verify the stability of the MR results.

### Statistical analysis

In this MR study, all the analyses were performed by the R software (version 4.2.0, http://www.R-project.org; The R Foundation, Vienna, Austria), with the "Two Sample MR" packages (version 0.5.6). All results were expressed as OR and its 95% CI, with $P < 0.05$ considered as statistically significant.

### Ethical considerations

This Mendelian randomization study used publicly available, anonymized genetic and phenotypic data. Since no individual-level data were collected and no human subjects were directly involved, ethical approval was not required.

## Results

### SNP dressing by screening and MR analysis

Through screening for SNPs, we obtained 117 LD-independent SNPs from the 25-hydroxyvitamin D GWAS data. We examined potential confounding factors associated with these 117 SNPs and confirmed that none were related. Therefore, these 117 SNPs could be utilized in the GWAS of ISSNHL. The F-statistics for the SNPs in this study all exceeded 10, reinforcing the reliability of the research findings. Detailed information can be found in S1 Appendix.

The casual impact on ISSNHL of each IV was shown in the forest plot (Fig 2). It can be seen from the red line at the bottom of the forest plot that the combined analysis result crosses over 0, indicating that the level of serum 25-hydroxyvitamin D has no causal effect on ISSNHL.

The findings from IVW, weighted median, weighted model, simple model, and MR-Egger regression analyses are presented in Table 1. In this investigation, the IVW method yielded the primary result concerning causal effects, which indicated that serum concentrations of 25-hydroxyvitamin D do not exert a causal effect on ISSNHL. (IVW; Odds Ratio = 1.09, 95% Confidence Interval = 0.81 to 1.48, P = 0.573). The analysis results of weighted median, weighted model, simple model, and MR-Egger regression were consistent with the IVW result (Fig 3).

The results of the heterogeneity analysis showed that there was no heterogeneity in the IVW analysis, which was evaluated by Cohran's Q test ($P > 0.05$, Table 2). The MR-Egger regression indicated that IVs have no pleiotropy of IVs (intercept = 0.002233724, se = 0.00670676, P = 0.739709 (> 0.05)). There was no outliers exist, and the funnel plot was roughly symmetrical (Fig 4A). The leave-one-out analysis was performed by systematically excluding individual SNPs and then assessing the cumulative effects of the remaining SNPs. The findings indicated that the results of the remaining SNPs consistently fell to the right of the invalid line following the exclusion of any single SNP. This observation substantiates that no individual SNP exerted a disproportionate influence on the MR results (Fig 4B), thereby affirming the robustness of the MR findings.

### Reverse MR analysis

Inverse MR analysis was performed with ISSNHL as an exposure factor and serum 25-hydroxyvitamin D levels an outcome. The findings from IVW, weighted median, weighted model, simple model, and MR-Egger regression analyses

**Table 1. Associations between the serum 25-hydroxyvitamin D level and risk of idiopathic sudden sensorineural hearing loss.**

| Method | OR (95%CI) | *P*-value |
|---|---|---|
| MR-Egger | 1.03 (0.64–1.64) | 0.911318142 |
| Weighted median | 1.20 (0.76–1.90) | 0.435863431 |
| Inverse variance weighted | 1.09 (0.81–1.48) | 0.573084181 |
| Simple mode | 0.76 (0.28–2.04) | 0.59006239 |
| Weighted mode | 1.10 (0.71–1.72) | 0.663077565 |

**Table 2. Heterogeneity testing of IVs for the serum 25-hydroxyvitamin D level.**

| Method | Q | Q_df | Q_pval |
|---|---|---|---|
| MR Egger | 99.49292 | 113 | 0.813851 |
| Inverse variance weighted | 99.60385 | 114 | 0.829485 |

indicated that there was no evidence of a causal relationship between ISSNHL and serum 25-hydroxyvitamin D levels (Table 3). In this investigation, the IVW method yielded the primary result concerning causal effects (IVW; Odds Ratio = 1.00, 95% Confidence Interval = 0.99 to 1.00, P = 0.282). Heterogeneity analysis showed that there was no heterogeneity in the IVW analysis, which was evaluated by Cohran's Q test ($P > 0.05$, Table 4). Pleiotropy test indicated that IVs have no pleiotropy of IVs (intercept = 0.00323687, se = 0.00233689, P = 0.30027999 (>0.05)). Detailed information can be found in S2 Appendix.

## Discussion

Numerous studies have established a correlation between serum vitamin D levels and ISSNHL. In a study conducted by Zandi et al. [11], a cohort comprising 50 patients diagnosed with ISSNHL and 50 healthy individuals without hearing impairment, serving as a control group, was analyzed. The findings indicated a heightened prevalence of vitamin D insufficiency among individuals with SSNHL, thereby suggesting a potential link between serum vitamin D levels and the onset of SSNHL. Additionally, Szeto et al. [24] conducted a cross-sectional analysis involving 1,123 participants aged 70 years and older, which revealed that low vitamin D status was correlated with low-frequency hearing loss (LFHL) and speech-frequency hearing loss (SFHL) in the elderly population. Furthermore, a survey by Chen et al. [25] investigated the relationship between serum concentrations of 25-hydroxyvitamins D2 and D3 and hearing loss in U.S. adults, uncovering a positive correlation between serum 25(OH)D2 levels and both LFHL and SFHL within the studied group. An L-shaped relationship was also identified between serum 25(OH)D3 and LFHL and SFHL, indicating that elevated serum 25(OH)D3 levels were associated with a reduced risk of high-frequency hearing loss (HFHL) in Chen's study.

The presence of a causal relationship between the two variables remains ambiguous. To address this uncertainty, we conducted a MR study aimed at exploring the potential causal association between them. This investigation was facilitated by the employment of publicly accessible GWAS databases, with meticulous attention paid to controlling for various confounding factors. Our findings suggest that there is no causal relationship between serum vitamin D levels and ISSNHL. This outcome stands in contrast to the findings of current observational studies. Based on this, a reverse MR analysis was further incorporated to investigate whether the occurrence of ISSNHL was correlated with the levels of serum vitamin D. Nevertheless, negative outcomes were still yielded. To the best of our knowledge, this study represents the inaugural effort to assess the causal relationship between these two variables. This result has significant clinical implications, as it suggests that the supplementation of vitamin D may not confer protective effects against sudden hearing loss. Clinicians should exercise caution when recommending vitamin D supplementation for this purpose, as it may lead to unnecessary healthcare expenditures without demonstrable benefit.

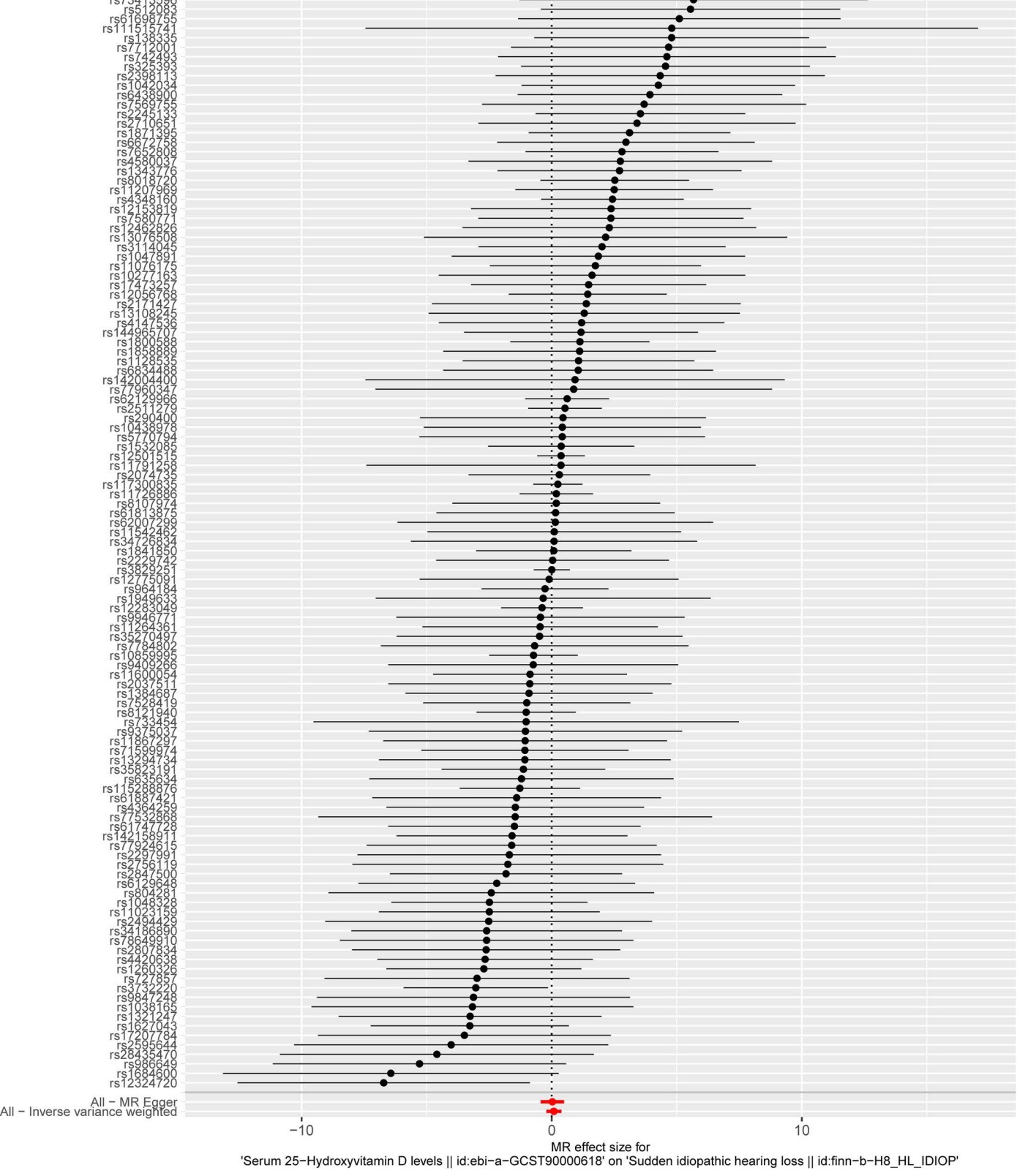

**Fig 2. The forest plots illustrating the estimated causal effects of individual SNPs.**

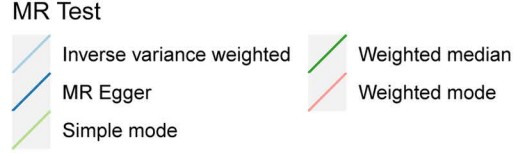

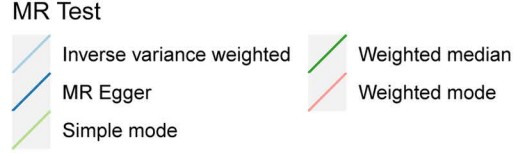

**Fig 3. Scatter plot of MR analysis.** The slope of each colored line represent the estimated MR effect in different models.

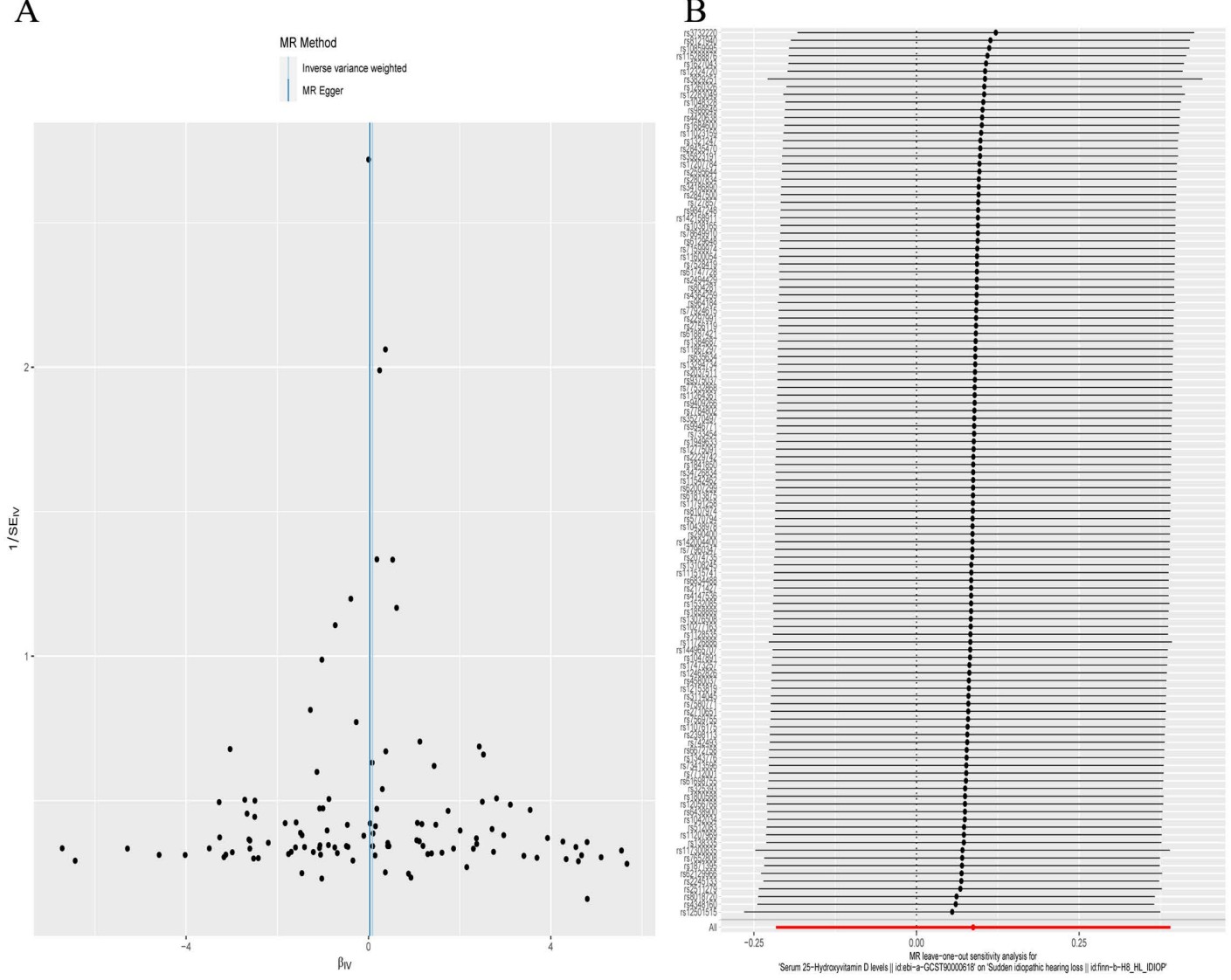

**Fig 4. Sensitivity analysis.** A. Funnel plot for the instrumental variables. B. Leave-one-out of sensitivity analysis. Calculate the MR results of the remaining IVs after removing the IVs one by one.

**Table 3. Associations between ISSNHL and the serum 25-hydroxyvitamin D level.**

| Method | OR (95%CI) | *P*-value |
|---|---|---|
| **MR-Egger** | 0.99 (0.98–1.00) | 0.225176688 |
| **Weighted median** | 1.00 (0.99–1.00) | 0.429928280 |
| **Inverse variance weighted** | 1.00 (0.99–1.00) | 0.281589127 |
| **Simple mode** | 1.00 (0.99–1.02) | 0.705586463 |
| **Weighted mode** | 0.99 (0.99–1.00) | 0.182769194 |

**Table 4. Heterogeneity analysis.**

| Method | Q | Q_df | Q_pval |
|---|---|---|---|
| MR Egger | 0.3308 | 2 | 0.84757 |
| Inverse variance weighted | 2.2493 | 4 | 0.52230 |

The observed discrepancies may be attributed to the inherent limitations associated with observational research methodologies. Primarily, these studies utilized cross-sectional data, which inherently restricts the ability to establish a causal relationship over time between serum 25-hydroxyvitamin D levels and ISSNHL. Furthermore, the focus of these investigations was confined to specific populations or to elderly individuals identified as being at heightened risk for hearing impairment. Additionally, the constraints of the available databases limited the capacity to control for all potential confounding variables, such as comorbid conditions like osteoporosis and fluctuations in parathyroid hormone levels, both of which may influence vitamin D levels and auditory health. Moreover, the findings may be affected by a bidirectional relationship, for instance, individuals with hearing impairments may tend to remain indoors more frequently, resulting in decreased sunlight exposure and, subsequently, vitamin D deficiency [26]. Another limitation of the studies in question is the lack of data regarding specific variables of interest, such as vitamin B-12 levels and the season during which 25-hydroxyvitamin D levels were assessed, both of which could potentially impact either hearing loss or vitamin D concentrations. Our MR study presents several advantages. The utilization of extensive genetic consortium data pertaining to serum 25-hydroxyvitamin D levels, which encompasses a considerable sample size (n = 33,996), in conjunction with ISSNHL risk data derived from a substantial number of cases (1,491) and controls (196,592), has facilitated a more rigorous examination of our hypothesis. This approach mitigates the limitations typically associated with individual-level data from studies with smaller cohorts. Furthermore, the application of stringent criteria for instrumental variables and the two-sample MR analysis has ensured the minimization of potential confounding factors.

In our MR analysis, we investigated the causal relationship between serum 25-hydroxyvitamin D concentrations and ISSNHL. One potential concern raised is the influence of seasonal variations on serum vitamin D levels, as sunlight exposure, a primary source of vitamin D synthesis, varies significantly across seasons. Seasonal fluctuations in vitamin D levels are well-documented, with higher concentrations typically observed during summer months due to increased ultraviolet B (UVB) radiation exposure, and lower levels during winter months. These variations could theoretically impact the estimation of vitamin D's causal effect on ISSNHL if not properly accounted for. However, a key strength of our MR approach is its robustness to such environmental confounders, including seasonal variations. MR leverages genetic variants as instrumental variables, which are randomly assigned at conception and are not influenced by seasonal or environmental factors. This methodology is predicated on the assumption that genetic variations influence health outcomes exclusively through designated exposure factors. The random assignment of IVs at conception effectively positions the MR analysis as a "natural randomized controlled trial," serving as a robust mechanism for inferring causal relationships. Since the genetic determinants of serum 25-hydroxyvitamin D levels are fixed and independent of temporal or seasonal changes, our MR estimates are less susceptible to bias from these fluctuations. This inherent feature of MR provides a more reliable estimate of the causal effect, even in the presence of seasonal variability in vitamin D levels [27].

It also should be noted that hearing loss manifests differently across various frequency ranges, including low, mid, and high frequencies, each potentially associated with distinct pathological mechanisms. Although our study did not stratify the analysis by frequency ranges, the findings provide valuable insights into the overall causal relationship between serum 25-hydroxyvitamin D concentrations and ISSNHL. ISSNHL typically involves multiple frequency ranges, and our results suggest that vitamin D levels may not exert a significant causal effect on the condition as a whole. However, future research should explore whether vitamin D levels differentially impact hearing loss across specific frequency ranges, as this could reveal more nuanced relationships and inform targeted interventions.

Our MR results estimate the LATE of serum 25-hydroxyvitamin D on ISSNHL risk, representing the causal effect among compliers whose vitamin D levels are modifiable by the genetic instruments. This subgroup-specific effect may not equate to the average causal effect in the general population, as genetic variants typically exert smaller physiological effects compared to clinical interventions. The monotonicity assumption underlying LATE, while supported by our selection of unidirectional vitamin D-associated SNPs from GWAS, remains untestable and warrants cautious interpretation. Notably, non-compliers and effect heterogeneity across subpopulations could limit generalizability. These considerations align with broader limitations of MR for estimating population-wide effects but do not invalidate the utility of LATE for identifying modifiable biological pathways. Future studies using vitamin D supplementation trials or gene-environment interaction analyses may help bridge this translational gap.

Emerging evidence highlights vitamin D as a key immunomodulator. As demonstrated by Murdaca et al. [28], 25(OH)D deficiency impairs the balance between pro-inflammatory and regulatory immune responses, particularly through suppression of Th17-mediated pathways and enhancement of Treg cell differentiation. Notably, the immunomodulatory effects of vitamin D may be partially mediated by gut microbiota. Murdaca et al. [29] revealed that vitamin D receptor signaling promotes colonization of anti-inflammatory commensal bacteria, while vitamin D deficiency exacerbates dysbiosis and gut barrier dysfunction. This 'gut-immune axis' could theoretically extend to the inner ear microenvironment, where microbial metabolites or systemic inflammation triggered by dysbiosis might precipitate cochlear inflammation and vascular compromise. Although our MR analysis focused on serum 25(OH)D, the observed association with ISSNHL might be amplified through microbiota-immune interactions. For instance, vitamin D sufficiency could maintain gut barrier integrity, thereby reducing bacterial translocation and subsequent endotoxin-induced cochlear microvascular inflammation - a hypothesized mechanism in ISSNHL pathogenesis.

While our research yielded noteworthy findings, it is crucial to recognize certain limitations. Firstly, the study's participant pool was predominantly composed of individuals of European descent, indicating the need for additional data collection and analysis to ascertain whether the results can be extrapolated to other ethnic groups. Secondly, hearing loss was operationally defined as a pure tone average exceeding 25 dB in either ear at low frequencies, specifically across 500, 1,000, and 2,000 Hz, as well as at higher frequencies of 4,000, 6,000, and 8,000 Hz. It is important to note that the frequency ranges utilized in this study were not exhaustively analyzed. This was due to the lack of detailed frequency-specific data in the datasets used for our MR analysis. As hearing loss can manifest differently across frequency ranges, future studies should aim to investigate whether serum 25-hydroxyvitamin D concentrations have frequency-specific effects on hearing loss, to provide deeper insights into the role of vitamin D in auditory health and help identify potential subgroups that may benefit from vitamin D supplementation. Thirdly, despite the robustness of our MR analysis to seasonal variations, we acknowledge that future studies could further explore the impact of seasonal changes on serum vitamin D levels and their potential association with ISSNHL. For instance, longitudinal studies measuring vitamin D levels across different seasons could provide additional insights into how temporal fluctuations might influence the risk of ISSNHL. Incorporating such data into future MR analyses or other causal inference frameworks could help refine our understanding of the relationship between vitamin D and hearing health. Fourthly, we acknowledge that conventional observational methods like OLS regression offer distinct advantages for probing nonlinear relationships and adjusting for measured confounders. However, these approaches are ill-suited to our research context for two reasons. First, the publicly available GWAS data lack individual-level measurements of critical covariates. Second, and more fundamentally, the bidirectional relationship between vitamin D status and hearing-related disability_whereby severe ISSNHL may limit sunlight exposure, creating reverse causality_renders OLS estimates vulnerable to substantial bias. Mendelian randomization provides a principled alternative by leveraging genetic variants as instrumental variables, naturally adjusting for lifelong confounding through Mendel's laws of inheritance. While this approach requires careful evaluation of pleiotropy, as addressed through sensitivity analyses, it avoids the insurmountable endogeneity problems inherent to observational vitamin D studies. Furthermore, the research underscores the significant role of genetic factors in the onset of ISSNHL and

vitamin D deficiency, suggesting a complex interplay of origins for ISSNHL. In other word, while we did not find evidence for a causal relationship between vitamin D and sudden hearing loss, it remains plausible that vitamin D could influence auditory function through mechanisms such as immune modulation or inflammatory response, which may not be captured in our analysis. Additionally, environmental and social factors, which are non-genetic in nature, also play a substantial role in these conditions, however, these elements were not comprehensively addressed in this study. Moreover, Due to the lack of individual-level data for the exposure and outcome samples, we are unable to assess potential overlaps between these samples which may leads to bias of our finding.

## Conclusion

Our MR-based investigation did not reveal any substantial evidence of a significant causal relationship between genetically determined serum 25-hydroxyvitamin D levels and the risk of ISSNHL among individuals of European ancestry. Nevertheless, there remains a pressing need for additional longitudinal studies employing larger datasets and more heterogeneous populations to investigate the causal mechanisms involved.

## Supporting information

**S1 Appendix. SNP dressing by screening.**
(CSV)

**S2 Appendix. Reverse MR analysis data.**
(ZIP)

**S1 Checklist. STROBE-MR-checklist-fillable20241027.**
(DOCX)

## Acknowledgments

The authors thank all the participants and support of the Central Laboratory members of The Affiliated Panyu Central Hospital, Guangzhou Medical University and member of Department of laboratory medcine, Third affiliated hospital of Sun Yat-sen University.

## Author contributions

**Data curation:** Xiaoming Li, Zeping Han.

**Investigation:** Ying Zhao.

**Resources:** Wenfeng Luo.

**Validation:** Fangmei Xie, Xiaoyu Song.

**Visualization:** Jian Shen.

**Writing – original draft:** Ying Zhao, Cong Yu.

**Writing – review & editing:** Hengchang Sun, Jinhua He.

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
