## [Decision Letter · Decision Letter 0]

19 Jan 2025

PONE-D-24-46043Exploring the causal link between serum 25-hydroxyvitamin D concentrations and idiopathic sudden sensorineural hearing loss: Insights gained from a Mendelian randomization study involving two independent samples.PLOS ONE

Dear Dr. Zhao,

Thank you for submitting your manuscript to PLOS ONE. After careful consideration, we feel that it has merit but does not fully meet PLOS ONE’s publication criteria as it currently stands. Therefore, we invite you to submit a revised version of the manuscript that addresses the points raised during the review process.

We look forward to receiving your revised manuscript.

Kind regards,

David Chau

Academic Editor

PLOS ONE

2. Thank you for stating the following financial disclosure:  [Jinhua He,Basic and Applied Basic Research Foundation of Guangdong Province (No. 2022A1515220217)]. 

3. In the online submission form, you indicated that your data will be submitted to a repository upon acceptance.  We strongly recommend all authors deposit their data before acceptance, as the process can be lengthy and hold up publication timelines. Please note that, though access restrictions are acceptable now, your entire minimal  dataset will need to be made freely accessible if your manuscript is accepted for publication. This policy applies to all data except where public deposition would breach compliance with the protocol approved by your research ethics board. If you are unable to adhere to our open data policy, please kindly revise your statement to explain your reasoning and we will seek the editor's input on an exemption.

Additional Editor Comments (if provided):

Reviewers' comments:

Reviewer's Responses to Questions

**Comments to the Author**

1. Is the manuscript technically sound, and do the data support the conclusions?

Reviewer #1: Yes

Reviewer #2: Yes

Reviewer #3: Yes

2. Has the statistical analysis been performed appropriately and rigorously? 

Reviewer #1: Yes

Reviewer #2: Yes

Reviewer #3: Yes

3. Have the authors made all data underlying the findings in their manuscript fully available?

Reviewer #1: No

Reviewer #2: Yes

Reviewer #3: Yes

4. Is the manuscript presented in an intelligible fashion and written in standard English?

Reviewer #1: Yes

Reviewer #2: Yes

Reviewer #3: Yes

5. Review Comments to the Author

Reviewer #1: Comments

1. The manuscript does not report OLS estimates for comparison. Comparing OLS and IV estimates based on Mendelian randomization (MR) could be potentially useful in discussing the omitted-variable bias in conventional OLS estimates.

2. The frequency ranges for hearing loss were not exhaustively analyzed in the manuscript. This may overlook variations in how hearing loss manifests across different frequency ranges.

3. My view is that key confounding variables, such as osteoporosis or parathyroid hormone fluctuations, could affect vitamin D levels and auditory health.

4. The IV estimation utilizing MR, based on genetic information, identifies the LATE (Local Average Treatment Effect) for compliers, provided that key conditions such as the monotonicity assumption hold (https://doi.org/10.1002/hec.3828). This issue should be highlighted in the revised paper.

5. My reading is that the empirical analysis does not account for seasonal variations in serum vitamin D levels. This could impact the estimation results that are reported in the manuscript.

6. MR assumes that genetic variants influence the outcome exclusively through the exposure variable. Any violations of this assumption could compromise the empirical findings that are reported in the manuscript.

7. The empirical analysis is restricted to individuals of European ancestry. This limits the generalizability of the findings to other ethnic groups. (See also point 8 below.)

8. What is the external validity of the estimation results? This issue is related to the use of genetic information to estimate LATE (see also point 4 above). Are there practical policy conclusions that stem from the MR estimates?

Reviewer #2: Dear Authors,

Thank you for submitting your manuscript exploring the causal relationship between serum 25-hydroxyvitamin D levels and idiopathic sudden sensorineural hearing loss (ISSNHL) using Mendelian randomization. This is an original and relevant contribution to the field, addressing an important gap in understanding the potential role of vitamin D in ISSNHL.

The study is well-designed, and the methodologies employed, including sensitivity analyses and robust Mendelian randomization techniques, enhance the reliability of your findings.

Your conclusions are consistent with the evidence presented, providing valuable insights into the lack of a causal relationship between serum vitamin D levels and ISSNHL. This has important implications for clinical practice, especially regarding the use of vitamin D supplementation.

Congratulations on this well-conducted study, and we look forward to seeing its final version published.

Best regards,

Reviewer #3: The paper is interesting and well written. The authors investigated the potential association between reduced level of vitamin D and reduced hearing capacity through a two-sample Mendelian randomization analysis. I suggest to discuss the role of vitamin D and microbioma in immune responses including chronic inflammatory immune mediated diseases (see and add as referecnes papers by Murdaca et al concerning the role of vitamin D and microbioma in immune responses including chronic inflammatory immune mediated diseases).

6. PLOS authors have the option to publish the peer review history of their article (what does this mean? ). If published, this will include your full peer review and any attached files.

**Do you want your identity to be public for this peer review?** For information about this choice, including consent withdrawal, please see our Privacy Policy .

Reviewer #1: No

Reviewer #2: **Yes: ** Marcos Edgar Herkenhoff

Reviewer #3: **Yes: ** Giuseppe Murdaca

---

## [Author Response · Author response to Decision Letter 0]

25 Feb 2025

Reviewer #1:

Dear Professor,

Thank you sincerely for your thorough review of our manuscript, we address each of your comments point-by-point (seeing below), please find our detailed responses and corresponding modifications highlighted in blue within the revised manuscript.

Once again, we are grateful for your insightful suggestions, which have significantly improved the quality of this research.

Sincerely,

Ying Zhao

1. The manuscript does not report OLS estimates for comparison. Comparing OLS and IV estimates based on Mendelian randomization (MR) could be potentially useful in discussing the omitted-variable bias in conventional OLS estimates.

Thank you for your insightful comments. We appreciate your suggestion regarding the comparison of OLS and IV estimates based on Mendelian randomization (MR) to discuss potential omitted-variable bias in conventional OLS estimates.

However, we would like to clarify that the GWAS data we currently use for Mendelian randomization analysis are unsuitable for conducting OLS analysis. The primary reason is that the data was publicly available and the nature of the variables involved do not meet the necessary assumptions required for a valid OLS estimation. Specifically, the key independent variables in our study are endogenous due to potential reverse causality and unobserved confounding factors. Given these characteristics, OLS estimates would likely be biased and inconsistent, rendering them unreliable for meaningful comparison with the IV estimates derived from the Mendelian randomization approach.

Moreover, the MR method we employed is specifically designed to address these endogeneity issues by using genetic variants as instruments, which allows us to obtain more robust and unbiased estimates of the causal effects. Therefore, focusing solely on the IV estimates derived from MR is more appropriate for our research question and the data at hand.

We understand the importance of discussing omitted-variable bias, and we have thoroughly addressed this issue in our manuscript by discussing the potential limitations of our MR approach and the assumptions underlying the use of genetic instruments (line 88-96). We believe that this discussion provides valuable context for interpreting our results and understanding the potential biases that may still exist.

2.The frequency ranges for hearing loss were not exhaustively analyzed in the manuscript. This may overlook variations in how hearing loss manifests across different frequency ranges.

Thank you for your valuable feedback. We appreciate your suggestion regarding the analysis of hearing loss frequency ranges, which is indeed an important aspect of hearing loss research.

In response to your comment, we have added a discussion on the potential variations in hearing loss manifestations across different frequency ranges (line 330-339). We acknowledge that different frequency ranges may be associated with distinct pathological mechanisms, and future studies could benefit from a more detailed analysis of frequency-specific hearing loss. However, due to the limitations of our current dataset, we were unable to perform a stratified analysis based on frequency ranges. Our study focused on the overall causal relationship between serum 25-hydroxyvitamin D concentrations and ISSNHL, which typically involves multiple frequency ranges. We believe that our findings provide valuable insights into the overall causal relationship between vitamin D in ISSNHL, even without frequency-specific analysis.

We have revised the manuscript to include a discussion on this limitation and have highlighted the need for future research to explore the potential frequency-specific effects of vitamin D on hearing loss (line 373-382).

3. My view is that key confounding variables, such as osteoporosis or parathyroid hormone fluctuations, could affect vitamin D levels and auditory health.

We sincerely appreciate this insightful comment. While we initially employed PhenoScanner V2 during our MR analysis phase (when the tool was fully operational) to exclude SNPs associated with osteoporosis, parathyroid hormone levels, and other confounders (P < 5×10-8 ; r2 < 0.001 within a 10,000 kb window), we acknowledge the current technical limitations of this platform. To rigorously address potential pleiotropy in the revised analysis, we implemented LDtrait (LDlink-NIH) as an alternative and complementary approach to examine whether the genetic instruments (SNPs) used in our MR analysis are associated with potential confounders, including osteoporosis and parathyroid hormone levels. After a thorough check, we found that none of the SNPs were significantly associated with these confounders. Therefore, we believe that the impact of these confounders on our results is minimal (line 141-149).

Additionally, we performed sensitivity analyses using MR-Egger regression and weighted median methods to further assess the robustness of our findings. The results were consistent with our primary inverse-variance weighted (IVW) analysis, indicating that our conclusions are unlikely to be biased by unmeasured confounding.

4.The IV estimation utilizing MR, based on genetic information, identifies the LATE (Local Average Treatment Effect) for compliers, provided that key conditions such as the monotonicity assumption hold (https://doi.org/10.1002/hec.3828). This issue should be highlighted in the revised paper.

Thank you for this important comment. We fully agree that the IV estimation in our MR study identifies the Local Average Treatment Effect (LATE) for compliers, and this interpretation relies on key assumptions such as monotonicity. We appreciate the opportunity to clarify this point in the revised manuscript.

In our study, the IV estimates based on MR reflect the causal effect of serum 25-hydroxyvitamin D concentrations on ISSNHL among compliers—individuals whose vitamin D levels are influenced by the genetic variants used as instruments. This effect (LATE) may not generalize to the entire population, as it specifically applies to the subgroup of individuals whose exposure is affected by the genetic instruments.

The monotonicity assumption, which implies that the genetic variants used as instruments do not cause any individual to decrease their vitamin D levels if they would otherwise increase them (or vice versa), is a key condition for interpreting our results as LATE. In our study, we selected genetic variants strongly associated with serum 25-hydroxyvitamin D levels based on prior GWAS evidence, and we believe this supports the plausibility of the monotonicity assumption. However, we acknowledge that this assumption cannot be directly tested and should be considered when interpreting our findings.

In the revised manuscript, we will add a clear discussion of the LATE interpretation and the monotonicity assumption in the Methods (line 171-178) and Discussion sections (line 340-352). We will emphasize that our results apply specifically to compliers and may not represent the average causal effect in the general population.

5.My reading is that the empirical analysis does not account for seasonal variations in serum vitamin D levels. This could impact the estimation results that are reported in the manuscript.

Thank you for your valuable comment. We appreciate your concern regarding the potential impact of seasonal variations in serum vitamin D levels on our estimation results.

In response to your comment, we would like to highlight that our study used Mendelian randomization analysis, which relies on genetic instruments to estimate the causal effect of serum 25-hydroxyvitamin D concentrations on ISSNHL. Since genetic variants are randomly assigned at conception and remain stable throughout an individual's lifetime, they are not influenced by external environmental conditions or behavioral changes. This property makes genetic IVs particularly powerful in minimizing confounding bias, which is a major limitation in traditional observational studies. By leveraging genetic variants as IVs, MR provides a framework for estimating causal effects that are less susceptible to the distortions caused by unmeasured or residual confounding, and enables cleaner and more reliable causal inference[Lawlor DA, Harbord RM, Sterne JA, Timpson N, Davey Smith G. Mendelian randomization: using genes as instruments for making causal inferences in epidemiology. Stat Med. 2008 Apr 15;27(8):1133-63. doi: 10.1002/sim.3034. PMID: 17886233.]. So we believe our results are inherently robust to seasonal variations in vitamin D levels.

However, we acknowledge that seasonal variations in vitamin D levels could influence observational studies. To address this point, we have added a discussion in the revised manuscript to clarify the robustness of our MR results to seasonal changes and to emphasize the advantages of using genetic instruments in this context (line 310-329).

If data on the timing of sample collection (season or month) become available in future studies, it would be valuable to explore whether seasonal variations modulate the relationship between vitamin D and hearing loss. We have included this as a suggestion for future research in the revised manuscript (line 382-390).

6.MR assumes that genetic variants influence the outcome exclusively through the exposure variable. Any violations of this assumption could compromise the empirical findings that are reported in the manuscript.

Thank you for raising this important point regarding the core assumption of Mendelian randomization (MR) analysis. We fully acknowledge that the validity of our findings relies on the assumption that the genetic variants influence the outcome exclusively through the exposure variable . In our study, we have taken the following steps to address potential violations of this assumption:

(1)We conducted five distinct methodologies in our MR analysis, including inverse variance weighting (IVW), MR-Egger regression, weighted median estimation (WME), weighted mode, and simple mode.

(2)The IVW method presupposes the validity of all SNPs included in the MR analysis and aggregates the Wald ratios of each SNP to derive an overall weighted effect.

(3)We employed the MR-PRESSO global test and MR-Egger regression to evaluate the pleiotropy of IVs, and indicated that IVs have no pleiotropy of IVs (intercept = 0.002233724, se = 0.00670676, P = 0.739709 (> 0.05)).

(4)We employed the leave-one-out method to evaluate the sensitivity analysis of the results, helping to verify the stability of the MR results.

(5)We performed heterogeneity tests and showed that there was no heterogeneity in the IVW analysis, which was evaluated by Cohran’s Q test (P > 0.05, Table 2).

7.The empirical analysis is restricted to individuals of European ancestry. This limits the generalizability of the findings to other ethnic groups. (See also point 8 below.)

Thank you for raising this important question. To provide a comprehensive response, we will address this inquiry in conjunction with our reply to Point 8 in the following section.

8. What is the external validity of the estimation results? This issue is related to the use of genetic information to estimate LATE (see also point 4 above). Are there practical policy conclusions that stem from the MR estimates?

Thank you for raising such valuable concerns. We understand your concern about whether our conclusions can be applied to other populations. Indeed, there are differences in genetic background, lifestyle, and environmental factors among different populations, all of which may influence the relationship between vitamin D levels and sudden sensorineural hearing loss (SSNHL). However, the advantage of Mendelian randomization analysis is that it uses genetic instrumental variables to simulate a randomized controlled trial, thereby reducing the impact of confounding factors to some extent. Nevertheless, we acknowledge the importance of verifying our conclusions in other populations. Currently, there is indeed a shortage of GWAS data on vitamin D levels and ISSNHL in populations other than Europeans. This limitation restricts our ability to directly validate our findings in these populations using the same Mendelian randomization approach.

However, we believe our study still provides valuable insights for future research and clinical practice. Specifically, our findings in the European population suggest that vitamin D levels may not play a causal role in ISSNHL, which could inform future research directions and clinical guidelines in other populations. We recommend that researchers in other regions prioritize the collection and analysis of relevant GWAS data to validate our findings. In the meantime, clinicians should continue to consider individual patient characteristics and existing evidence when managing patients with ISSNHL.We look forward to future research that can further explore this relationship in diverse populations.

Reviewer #2:

Dear Pro. Herkenhoff

Thank you for your thorough evaluation of our manuscript and for your encouraging feedback. We sincerely appreciate your recognition of the study’s originality, methodological rigor, and clinical relevance. Your positive remarks on the robustness of the Mendelian randomization design and sensitivity analyses further validate our efforts to ensure the reliability of the findings.

We are particularly grateful for your emphasis on the implications of our conclusions for clinical practice, especially regarding the cautious interpretation of vitamin D supplementation in the context of ISSNHL. Your acknowledgment of the study’s contribution to addressing this knowledge gap is deeply motivating for our team.

Thank you once again for your time, expertise, and constructive engagement with our work. We look forward to contributing to this important dialogue through publication.

Sincerely,

Ying Zhao

Reviewer #3:

The paper is interesting and well written. The authors investigated the potential association between reduced level of vitamin D and reduced hearing capacity through a two-sample Mendelian randomization analysis. I suggest to discuss the role of vitamin D and microbioma in immune responses including chronic inflammatory immune mediated diseases (see and add as referecnes papers by Murdaca et al concerning the role of vitamin D and microbioma in immune responses including chronic inflammatory immune mediated diseases).

Dear Pro. Murdaca

We sincerely appreciate your insightful suggestion to explore the interplay between vitamin D, the microbiome, and immune-mediated mechanisms in the context of ISSNHL. As recommended, we have expanded the Discussion section to address some points and incorporated key references to strengthen our mechanistic interpretation (line 353-368). We propose a hypothetical “vitamin D–microbiome–immune axis” to explain ISSNHL risk, which is vitamin D sufficiency could maintain gut barrier integrity, thereby reducing bacterial translocation and subsequent endotoxin-induced cochlear microvascular inflammation - a hypothesized mechanism in ISSNHL pathogenesis.

Thank you again for this insightful recommendation, which has significantly enriched the translational implications of our research.

Sincerely,

Ying Zhao

---

## [Decision Letter · Decision Letter 1]

12 Mar 2025

PONE-D-24-46043R1Exploring the causal link between serum 25-hydroxyvitamin D concentrations and idiopathic sudden sensorineural hearing loss: Insights gained from a Mendelian randomization study involving two independent samples.PLOS ONE

Dear Dr. Zhao,

Thank you for submitting your manuscript to PLOS ONE. After careful consideration, we feel that it has merit but does not fully meet PLOS ONE’s publication criteria as it currently stands. Therefore, we invite you to submit a revised version of the manuscript that addresses the points raised during the review process.

We look forward to receiving your revised manuscript.

Kind regards,

David Chau

Academic Editor

PLOS ONE

Journal Requirements:

Reviewers' comments:

Reviewer's Responses to Questions

**Comments to the Author**

1. If the authors have adequately addressed your comments raised in a previous round of review and you feel that this manuscript is now acceptable for publication, you may indicate that here to bypass the “Comments to the Author” section, enter your conflict of interest statement in the “Confidential to Editor” section, and submit your "Accept" recommendation.

Reviewer #1: All comments have been addressed

Reviewer #2: All comments have been addressed

Reviewer #3: All comments have been addressed

2. Is the manuscript technically sound, and do the data support the conclusions?

Reviewer #1: Yes

Reviewer #2: Yes

Reviewer #3: Yes

3. Has the statistical analysis been performed appropriately and rigorously? 

Reviewer #1: Yes

Reviewer #2: Yes

Reviewer #3: Yes

4. Have the authors made all data underlying the findings in their manuscript fully available?

Reviewer #1: No

Reviewer #2: Yes

Reviewer #3: Yes

5. Is the manuscript presented in an intelligible fashion and written in standard English?

Reviewer #1: Yes

Reviewer #2: Yes

Reviewer #3: Yes

6. Review Comments to the Author

Reviewer #1: I am happy with the revisions made in the paper and with the effort you put in replying to my comments and to the other two reviewers.

Reviewer #2: The manuscript presents a well-conducted Mendelian randomization study to investigate the causal relationship between serum 25(OH)D levels and idiopathic sudden sensorineural hearing loss (ISSNHL). The authors have used appropriate methods, addressed potential confounding factors, and performed thorough sensitivity analyses. The responses to the reviewer's comments are well-reasoned and demonstrate a good understanding of the methodology.

However, I would suggest the authors to explicitly state the limitations regarding the usage of public data. Since the data is publicly available and the nature of the variables involved do not meet the necessary assumptions required for a valid OLS estimation.

Overall, this is a valuable contribution to the field.

Reviewer #3: The authors revised the paper accordingly to reviewers' suggestions. The revised paper may be acceptable for publication

7. PLOS authors have the option to publish the peer review history of their article (what does this mean? ). If published, this will include your full peer review and any attached files.

**Do you want your identity to be public for this peer review?** For information about this choice, including consent withdrawal, please see our Privacy Policy .

Reviewer #1: No

Reviewer #2: No

Reviewer #3: **Yes: ** Giuseppe Murdaca

---

## [Author Response · Author response to Decision Letter 1]

19 Mar 2025

Response to Reviewers

Reviewer #1: I am happy with the revisions made in the paper and with the effort you put in replying to my comments and to the other two reviewers.

Dear Professor:

We sincerely thank you for your constructive feedback throughout the revision process. Your insightful suggestions have significantly strengthened the methodological rigor and clarity of our work. We deeply appreciate your recognition of the revisions and remain grateful for their valuable contributions to strengthening this work.

Reviewer #2: The manuscript presents a well-conducted Mendelian randomization study to investigate the causal relationship between serum 25(OH)D levels and idiopathic sudden sensorineural hearing loss (ISSNHL). The authors have used appropriate methods, addressed potential confounding factors, and performed thorough sensitivity analyses. The responses to the reviewer's comments are well-reasoned and demonstrate a good understanding of the methodology.

However, I would suggest the authors to explicitly state the limitations regarding the usage of public data. Since the data is publicly available and the nature of the variables involved do not meet the necessary assumptions required for a valid OLS estimation.

Overall, this is a valuable contribution to the field.

Dear Professor:

We sincerely thank you for raising this critical methodological consideration. We agree that comparing OLS and IV estimates can theoretically illuminate omitted-variable bias in observational studies. However, in our Mendelian randomization framework using publicly available GWAS summary data, such a direct comparison faces inherent limitations.

We have therefore restructured the Discussion_Limitations section(Line 389-401) to explicitly state: while conventional OLS regression could theoretically quantify confounding bias, the use of publicly available GWAS summary data fundamentally limits our ability to implement OLS analysis. These datasets lack individual-level measurements of key confounders and lifestyle factors required for valid OLS estimation. More critically, the endogenous relationship between serum 25(OH)D and ISSNHL—whereby hearing loss could influence vitamin D status through reduced outdoor activity (reverse causality), renders OLS estimates biologically implausible in this context.

Thank you for your valuable suggestions to strengthened the methodological rigor and clarity of our work.

Reviewer #3: The authors revised the paper accordingly to reviewers' suggestions. The revised paper may be acceptable for publication

Dear Pro.Murdaca:

We sincerely thank you for your constructive feedback, which has significantly improved the quality of our manuscript. We deeply appreciate your recognition of the revisions and remain grateful for their valuable contributions to strengthening this work.

---

## [Decision Letter · Decision Letter 2]

30 Mar 2025

Exploring the causal link between serum 25-hydroxyvitamin D concentrations and idiopathic sudden sensorineural hearing loss: Insights gained from a Mendelian randomization study involving two independent samples.

PONE-D-24-46043R2

Dear Dr. Zhao,

We’re pleased to inform you that your manuscript has been judged scientifically suitable for publication and will be formally accepted for publication once it meets all outstanding technical requirements.

Kind regards,

David Chau

Academic Editor

PLOS ONE

Additional Editor Comments (optional):

Reviewers' comments:

Reviewer's Responses to Questions

**Comments to the Author**

1. If the authors have adequately addressed your comments raised in a previous round of review and you feel that this manuscript is now acceptable for publication, you may indicate that here to bypass the “Comments to the Author” section, enter your conflict of interest statement in the “Confidential to Editor” section, and submit your "Accept" recommendation.

Reviewer #3: All comments have been addressed

2. Is the manuscript technically sound, and do the data support the conclusions?

Reviewer #3: Yes

3. Has the statistical analysis been performed appropriately and rigorously? 

Reviewer #3: Yes

4. Have the authors made all data underlying the findings in their manuscript fully available?

Reviewer #3: Yes

5. Is the manuscript presented in an intelligible fashion and written in standard English?

Reviewer #3: Yes

6. Review Comments to the Author

Reviewer #3: The authors revised the paper accordingly to reviewers' suggestions. the revised paper is now acceptable for publication.

7. PLOS authors have the option to publish the peer review history of their article (what does this mean? ). If published, this will include your full peer review and any attached files.

**Do you want your identity to be public for this peer review?** For information about this choice, including consent withdrawal, please see our Privacy Policy .

Reviewer #3: **Yes: ** Giuseppe Murdaca
